# Design, Characterization, and Preliminary Assessment of a Two-Degree-of-Freedom Powered Ankle–Foot Prosthesis

**DOI:** 10.3390/biomimetics9020076

**Published:** 2024-01-26

**Authors:** Tsung-Han Hsieh, Hyungeun Song, Tony Shu, Junqing Qiao, Seong Ho Yeon, Matthew Carney, Luke Mooney, Jean-François Duval, Hugh Herr

**Affiliations:** 1K. Lisa Yang Center for Bionics, Massachusetts Institute of Technology, Cambridge, MA 02142, USA; 2Department of Media Arts and Sciences, Massachusetts Institute of Technology, Cambridge, MA 02142, USA; 3Dephy Inc., 80 Central St, Suite 125, Boxborough, MA 01719, USA

**Keywords:** powered ankle prostheses, two-degrees-of-freedom, subtalar joint, transtibial amputation, eversion and inversion, level-ground walking, frontal plane motion, torque control

## Abstract

Powered ankle prostheses have been proven to improve the walking economy of people with transtibial amputation. All commercial powered ankle prostheses that are currently available can only perform one-degree-of-freedom motion in a limited range. However, studies have shown that the frontal plane motion during ambulation is associated with balancing. In addition, as more advanced neural interfaces have become available for people with amputation, it is possible to fully recover ankle function by combining neural signals and a robotic ankle. Accordingly, there is a need for a powered ankle prosthesis that can have active control on not only plantarflexion and dorsiflexion but also eversion and inversion. We designed, built, and evaluated a two-degree-of-freedom (2-DoF) powered ankle–foot prosthesis that is untethered and can support level-ground walking. Benchtop tests were conducted to characterize the dynamics of the system. Walking trials were performed with a 77 kg subject that has unilateral transtibial amputation to evaluate system performance under realistic conditions. Benchtop tests demonstrated a step response rise time of less than 50 milliseconds for a torque of 40 N·m on each actuator. The closed-loop torque bandwidth of the actuator is 9.74 Hz. Walking trials demonstrated torque tracking errors (root mean square) of less than 7 N·m. These results suggested that the device can perform adequate torque control and support level-ground walking. This prosthesis can serve as a platform for studying biomechanics related to balance and has the possibility of further recovering the biological function of the ankle–subtalar–foot complex beyond the existing powered ankles.

## 1. Introduction

Over the past decade, remarkable advancements have been achieved in bionic prostheses. These bionic prostheses, essentially powered robotic devices operated by microcontrollers, extend the capability of restoring biological functions beyond what conventional prostheses can offer. Although they are generally heavier than traditional passive-elastic prostheses, powered ankle-foot prostheses have demonstrated effectiveness in improving the walking economy of people with transtibial amputation [1,2,3]. This progress highlights a significant shift towards more advanced, functional prosthetic designs. Currently, all commercially available powered ankle-foot prostheses, including the Ottobock Empower [4] and Össur Proprio [5], are limited to one-degree-of-freedom (1-DoF) movements in the sagittal plane, specifically plantarflexion and dorsiflexion. However, research has highlighted the significance of frontal plane movements, such as eversion and inversion, in maintaining balance. Studies involving people without amputation have demonstrated that the ankle actively controls movements in the frontal plane to adjust the center of pressure and ensure balance during walking [6,7]. Additionally, recent advancements in peripheral neural interfaces have shown promise in preserving proprioception following lower limb amputation, leading to more robust and consistent motor control [8]. This progress, combined with the right robotic hardware, could allow for the complete restoration of the ankle-subtalar-foot complex’s biological function. These developments highlight the need for a powered 2-DoF ankle-foot prosthesis that not only is untethered but also approximates the function of a biological ankle-subtalar-foot complex. It should possess active control in both the sagittal and frontal planes. Crucially, the prosthesis must be capable of generating sufficient positive net work to support walking comfortably.

Several attempts have been made to develop a powered 2-DoF ankle-foot prosthesis. Bellman et al. proposed a mechanical design for a high-power 2-DoF ankle for performing tasks equivalent to a human ankle [9]. However, the design presents certain limitations. With a large size and weight, it poses a challenge for practical application. Ficanha et al. achieved a 2-DoF ankle prototype that is cable-driven, with electronics separate from the system. The device was able to modulate the impedance in both sagittal and frontal planes [10]. Kim et al. achieved a 2-DoF prosthesis emulator that is cable-driven and can control plantarflexion and inversion-eversion torque independently [11]. Clinical trials were also performed using the same system to examine frontal plane control on balance [12]. This system serves as a valuable tool in the study of biomechanics and in the search for better designs by emulating different control strategies. However, this design’s tethered nature restricts its use, limiting users to walking only on a treadmill. Jang et al. recently presented a self-contained 2-DoF ankle-foot prosthesis, noted for its lightweight design and suitability for walking on uneven terrain, that showcases innovations like a parallel linkage mechanism, a unidirectional parallel elastic actuator for energy storage, and capacitive force sensors for enhanced agility [13]. While this prosthesis shows promise, it is worth noting for context that testing on subjects with amputation has not yet been conducted, which will be a valuable step in further understanding its practical application.

This research focuses on creating and assessing an untethered, self-contained, powered 2-DoF ankle prosthesis, designed for level-ground walking and potentially assisting in stair navigation. The project encompasses the integration of a mechanical design that ensures controlled movement in both frontal and sagittal planes, along with a robust transmission capable of bearing body weight and facilitating significant push-off torque. It also includes the development of a compact electronic system, featuring a high-energy, high-output battery. The control system, aimed at replicating natural human walking dynamics, was tested on a subject with unilateral amputation for level-ground walking. This paper delves into the mechanical design, electronic systems, control strategies, and the prosthesis’s performance, offering insights and directions for future research.

## 2. Materials and Methods

### 2.1. Mechanical Design

The overview of the ankle prosthesis, as depicted in Figure 1, encapsulates the essence of our design objective. Our aim was to engineer a 2-DoF ankle prosthesis that functions without tethering, effectively supporting activities such as level-ground walking and maneuvering over uneven terrains. The design of this prosthesis was specifically tailored to support individuals weighing around 75 kg. To inform the design and ensure its suitability for this weight range, locomotion data from existing literature were utilized [14,15,16]. These sources provided valuable insights into the biomechanics of human gait, crucial for the prosthesis’s effective performance. The design specifications of the 2-DoF ankle prosthesis, which carefully strikes a balance between functionality and practicality, is outlined in Table 1.

Table 1 lists essential aspects of the 2-DoF ankle prosthesis, including its weight with and without the battery, dimensions, and the constrained degrees of movement in different planes. The table further elaborates on key technical details like variable transmission ratio, peak torques, battery voltage, and motor specifications. Notably, the transmission ratio varies due to the mechanism’s design, which is discussed in a subsequent section. The prosthesis’s range of motion is intentionally less than that of a biological ankle, a limitation influenced by size constraints. For this design, increasing the range of motion would result in a significantly larger and heavier robot, compromising practicality. While the prosthesis’s motion range is less than a biological ankle due to size limitations, it is sufficient for walking and assists in push-off during stair ascent, balancing functionality with practical constraints.

The actuator design of the ankle prosthesis employs two brushless electric motors (U8 Lite KV85, T-motor, Nanchang, Jiangxi, China), creating a differential drive system. This setup enables distinct movements: synchronized motor action results in sagittal plane motion, while asynchronous motor action facilitates coupled motion in both the frontal and sagittal planes. Each motor is linked to a two-stage belt drive, connected to a four-bar linkage that actuates the foot. PowerGrip GT3 timing belts (Stock Drive Products/Sterling Instrument, Hicksville, NY, USA) were utilized for both stages. The first stage features belts with a 3 mm pitch, 15 mm width, and 73 teeth, while the second stage employs belts with a 5 mm pitch, 15 mm width, and 40 teeth. Belt tensioners were incorporated not only to maintain appropriate belt tension for effective power transmission but also to simplify the assembly process. The essential elements of this transmission mechanism are depicted in Figure 2, illustrating the design that underpins the prosthesis’s movement capabilities.

The four-bar linkage incorporated in the transmission design is a crank-rocker type, resulting in a variable transmission ratio. Figure 3a illustrates a detailed representation of the four-bar linkage depicted in Figure 2. A planar four-bar linkage is a well-studied mechanism [17]. The transmission ratio, or mechanical advantage, from the input crank torque $T_{in}$ to output rocker torque $T_{out}$ can be solved via the principle of virtual work:(1)$$T_{in}\dot{\theta }=T_{out}\dot{\psi }$$
where
(2)$$R=\frac{T_{out}}{T_{in}}=\frac{\dot{\theta }}{\dot{\psi }}=\frac{\left(-ab\cos\psi \sin\theta -bg\sin\psi +ab\cos\theta \sin\psi \right)}{\left(ag\sin\theta +ab\cos\psi \sin\theta -ab\cos\theta \sin\psi \right)}.$$

Using Equation (Equation 2), the overall transmission ratio (including the contribution from the belt drive) versus the ankle angle in the sagittal plane can be derived, as illustrated in Figure 3b. Note that, as the joint moves toward the limit positions, the transmission ratio reaches infinity; this is known as the “toggle positions” in the four-bar mechanism [17]. At the toggle positions, the ankle is not back-drivable. There are 3D-printed Nylon (PA11) hard stops at the toggle positions to prevent the crank from moving beyond those toggle positions. The hard stops provide a 10° cushion for the cranks beyond the toggle positions before they would hit the main aluminum structure. Software hard stops were also implemented as unidirectional springs. Restricting the crank motion in the desired range (180°) simplifies solving the forward kinematics, as there exists a one-to-one relationship between crank and rocker angles. In other words, if the crank is able to pass the toggle positions, then the same rocker position can result in two different crank positions, making it difficult to map the forward kinematics.

### 2.2. Force and Torque Sensing

Full-bridge strain gauges (MMF307425, Micro-Measurements, Raleigh, NC, USA) were installed to the “tendon” parts, the connectors in the four-bar linkage of the ankle prosthesis, to measure force. The tendons were machined in 7075-T6 Aluminum. Each side of the tendon has a half-bridge configuration. This setup is depicted in Figure 4. This full-bridge configuration was selected because it can measure axial strain and has much better sensitivity than half-bridge and quarter-bridge configurations. In addition, this full-bridge configuration compensates for the Poisson effect and minimizes the effects of temperature. The joint torques are then derived using these force measurements in conjunction with the moment arms.

The performance of strain gauges is greatly affected by the installation process and packaging. To ensure proper installation, an installation kit (MMF307425, Micro-Measurements, Raleigh, NC, USA) from the vendor was used. Figure 5 displays the finished part.

As illustrated in Figure 5a, the strain gauges have very thin (32 AWG) pre-attached ribbon leads to minimize strain field interference. However, these leads are vulnerable in dynamic environments. To protect them, non-conductive, self-leveling silicone rubber with high elasticity was applied (3140 RTV, Dow Corning, Dow Way Midland, MI, USA). Additionally, protective covers were 3D printed to encase the strain gauges, and the wires were shielded in metal-braided sleeves and grounded to the microcontrollers, as depicted in Figure 5b.

Post-packaging, the sensors underwent calibration using a universal testing machine (5984, INSTRON, Norwood, MA, USA). To facilitate this, custom fixtures, machined from 304 stainless steel for their high strength, were used to secure the parts during testing. This ensured that the fixtures would not yield before the test components. Figure 6 illustrates both the design of these fixtures and the setup used for testing.

During the test, the sensors were subjected to both tension and compression in 200 N increments, culminating at a maximum force of 2000 N. The sensor data were captured by a control board (FlexSEA-Rigid, Dephy Inc., Boxborough, MA, USA), equipped with a cascaded RC low-pass filter (1.6 kHz cutoff) and a dual-stage instrumentation amplifier with high common-mode rejection ratio (CMRR) and a gain of 202.6. Figure 7 presents the test results and curve fitting, indicating a near-linear correlation between the applied external forces and the sensor readings.

For the sagittal plane, the moment arm is a function of the crank angle in the four-bar linkage. In Figure 8, $\theta_{0}$ represents the crank angle when the rocker (foot) is at the zero position (101.73°), and ϕ represents the angle relative to that zero position. To find the moment arm *l*, first calculate *c* using the law of cosines:(3)$$c=\sqrt{a^{2}+g^{2}-2ag\cos\left(\theta_{0}+\phi \right)}.$$

Then, set $s=\left(c+h+b\right)/2$. Using Heron’s formula,
(4)$$l=\left(2\sqrt{s(s-a)(s-b)(s-c)}\right)/h.$$

For frontal plane torque, even at the maximum eversion and inversion positions, the connector linkage’s motion on the frontal plane is negligible. Therefore, the moment arm is approximated as a constant (12.75 mm).

### 2.3. Kinematics Model

A kinematics model of the ankle was constructed in a virtual environment using MATLAB/Simulink Simscape Multibody (MathWorks Inc., Natick, MA, USA), as illustrated in Figure 9. The model simulates an RR-2RSS mechanism, where R represents a revolute joint, and S represents a spherical joint. A universal joint is formed by two revolute joints with mutually orthogonal axes and coincident origins. The world frame is a motionless, orthogonal, right-handed frame and serves as the ground of all the frame networks in the model. The spherical joints model the swivel bearings in the ankle, which have a 27° range of motion on the z-axis (hence a 13.5° inversion/eversion).

As mentioned, the crank motion in the four-bar linkage is constrained by hard stops, providing a one-to-one relationship between the crank and rocker position. To determine the sagittal plane joint angle, the kinematics of a 2D four-bar linkage was first simulated and fitted using the fit() command in MATLAB. The curve fitting was performed by non-linear least squares and a two-term Fourier series as base, which has the following form:(5)$$r\left(\theta \right)=a_{0}+a_{1}\cos\left(w\theta \right)+b_{1}\sin\left(w\theta \right)+a_{2}\cos\left(2w\theta \right)+b_{2}\sin\left(2w\theta \right),$$
where r(θ) is the rocker angle, and θ is the crank angle. The parameters $a_{0}$, $a_{1}$, $a_{2}$, $b_{1}$, $b_{2}$, and *w* were determined by the curve-fitting function. The fitted curve has a root-mean-square error (RMSE) of 0.009 degrees. Note that this works only for a 2D four-bar linkage. When the two motors are not moving in sync, the ankle performs coupled frontal and sagittal plane motion. However, the sagittal plane angle Φ can be calculated by the average of the two rockers: $\Phi =\left[r_{1}\left(\theta_{1}\right)+r_{2}\left(\theta_{2}\right)\right]/2$. Frontal plane motion, determined based on the difference in angles between crank 1 and crank 2, was also accurately characterized using the same method, with an RMSE of 6.446 × $10^{-5}$ degrees.

### 2.4. Electronics and Embedded System

Figure 10 depicts the electronic system of the 2-DoF ankle, powered by a 6-cell, 22 V Li-Po battery. The system includes a custom power distribution board featuring a step-down voltage regulator (D24V50F5, Pololu, Las Vegas, NV, USA), which converts 22 V to 5 V (5 A maximum) to power the ODROID-XU4 single-board computer (HARDKERNAL, AnYang, GyeongGi, Republic of Korea). The ODROID-XU4 runs high-level control algorithms and communicates with two FlexSEA-rigid boards (Dephy Inc., Boxborough, MA, USA) using a serial bus. FlexSEA-rigid boards are mid-level and low-level controllers. They process sensor data, including readings from strain gauges and encoders, and execute control over joint position and torque. A 14-bit absolute motor encoder (AS5047P, ams AG, Premstaetten, Austria) is mounted on each motor to sense motor position. A heavy-duty emergency stop (ED250, Albright International, Hampshire, UK) can be connected to the power board for emergency shut down. This emergency stop is optional and can be taken off and replaced by a shorted connector. Figure 11 illustrates a physical model of each component in the electronics system.

The ODROID-XU4 single-board computer is a computing device with more powerful, more energy-efficient hardware and a smaller form factor compared to the popular Raspberry Pi single-board computer [18]. The ODROID-XU4 on the 2-DoF ankle runs the Ubuntu operating system with a Linux real-time kernel. The communication stack utilizes a multi-threading technique and can communicate with each FlexSEA-rigid board at 2000 Hz. Sensor data from the powered ankle-foot prosthesis, such as timestamps, motor currents, encoder readings, joint torques, joint angles, moment arms, and other relevant metrics, were streamed wirelessly to a PC via WiFi. A graphical user interface (GUI) program running on the PC enabled visualization and recording of the streaming data for analysis and control development purposes. The wireless data transmission and GUI program allowed for rapid prototyping and iteration of control strategies by providing insight into the prosthesis state and dynamics during experiments.

The FlexSEA-rigid board is a specialized brushless drive and motor controller, specifically developed for use in wearable robotic applications [19]. FlexSEA-rigid is a single-package solution that integrates four logical blocks called Regulate, Manage, Execute, and Communicate. The Manage module is based on a powerful microcontroller (Cortex-M4, ARM, Cambridge, UK). It hosts mid-level control algorithms, which can be programmed by the developers. The Execute module, focused on brushless motor control, runs low-level controllers in response to commands from the Manage module. In the brushless drive, feedback loops close at 1 kHz for position and 10 kHz for current. The Regulate module oversees the Manage and Execute modules, encompassing on/off logic and sensors for monitoring experiments and the board’s state. This system ensures precise control and real-time supervision of the prosthesis’s functions. The Communicate module groups all of the communication peripherals in FlexSEA-rigid. Wired interfaces include one USB port and one RS-485 interface. Wireless capability can be achieved by connecting a pre-certified Bluetooth module (RN-42, Microchip, Chandler, AZ, USA). Additional peripherals such as strain gauges can be connected via the expansion connector on the board.

A custom battery module was constructed and integrated into the 2-DoF ankle. The battery case design, 3D printed in Nylon (PA 11 Black), incorporates a spring-loaded button, allowing the battery pack to securely latch inside the battery box. The angle of the battery box is adjustable to prevent it from hitting the shank of the wearer. Custom circuit boards were specifically designed and fabricated for the battery module. These include a power adapter board, which facilitates the transmission of both battery power and individual cell voltages through a connector. Additionally, there is a power distribution board, engineered to convert the battery’s 22 V down to 5 V, thereby supplying power to the ODROID-XU4. Additionally, the 22 V power is directed to two FlexSEA-rigid boards through an emergency stop connector. When the emergency stop is not in use, the connector is shorted. In addition to distributing power and converting voltage, there is also a timing circuit in the power distribution board to ensure that the battery can power the two FleaSEA-rigid boards and the ODROID-XU4 computer at the same time with a single-pole, single-throw rocker switch.

### 2.5. Control System Design and Evaluation

On each FlexSEA-rigid board, a proportional-derivative (PD) controller with a motor feedforward model was implemented for torque control, running at 1 kHz and using torque feedback. The process begins with converting the strain gauge signal to axial force, then to joint torque using the moment arm. The feedforward model produces a current command based on the desired torque, while the PD controller regulates motor current based on error signals and control gains. For controller testing, the ankle was rigidly fixed in a test fixture, and a separate fixture was created for free-space motion testing, as shown in Figure 12. To evaluate the torque controller performance, step response and torque bandwidth tests were conducted. For the bandwidth tests, a chirp signal ranging from 0.1 to 40 Hz over 80 s, oscillating between 4 and 40 N·m, was used as the desired torque command. This sweeping sine approach enabled a comprehensive evaluation of the controller’s torque tracking capabilities across a wide frequency range. For the step response tests, a desired step change in torque of 40 N·m was commanded. Analyzing the step response provided insight into the torque controller’s transient behavior. Together, the bandwidth and step response testing quantified both the dynamic and steady-state torque tracking accuracy, providing a thorough characterization of the torque control loop across operating conditions.

To achieve human-like walking dynamics for the prosthesis, an impedance control strategy was implemented on top of the underlying torque controller. Impedance control was first proposed by Hogan [20], who suggested representing the required joint torque as a series of piecewise impedance functions. Furthermore, Sup et al. performed a regression analysis on human gait data to characterize the joint impedance model as follows [21]:(6)$$\tau =k_{1}\left(\theta -\theta_{e}\right)+k_{2}{\left(\theta -\theta_{e}\right)}^{3}+b\dot{\theta },$$
where $k_{1}$ and $k_{2}$ characterize the linear and cubic stiffness, respectively, $\theta_{e}$ is the equilibrium angle, *b* is the linear damping coefficient, θ is the joint angle, and τ is the joint torque. The walking gait is further divided into three states, and each state has its own $k_{1}$, $k_{2}$, and $\theta_{e}$ values. A state machine was implemented on the ODROID-XU4 computer. The evaluation aimed to assess the prosthesis’s torque tracking performance under body weight and the mechanical transmission’s robustness. For simplicity, the impedance parameters on both motor drives were set identically. This meant the control focused on modeling ankle dynamics in the sagittal plane, while in the frontal plane, the ankle operated as a variable spring and damper, without emulating biological characteristics. This approach streamlined the testing process, focusing on key performance aspects. To evaluate torque tracking performance, the RMSE between the desired and measured torque of the entire stance period was calculated.

A 45-year-old female with a unilateral below-knee amputation was recruited to participate in a walking test. The subject has lived with the amputation for two years, stands 1.65 m tall, and weighs 77 kg. As a K4 level ambulator, the subject is proficient with standard passive prostheses and capable of walking at variable cadences. The walking test was conducted on level ground at a speed chosen by the subject. The impedance parameters of the prosthesis were manually tuned until the subject felt comfortable, focusing on achieving an optimal balance between device performance and user comfort. The entire setup can be viewed in Figure 13. The mechanical transmission’s robustness was assessed by monitoring for any occurrences of jump tooth on the pulleys, as indicated by the readings from the absolute encoders on the motors. This approach provided a direct evaluation of the transmission’s reliability under operational conditions.

## 3. Results

### 3.1. Integration of Dual Two-Stage Belt Drives and Four-Bar Linkage in Compact Design

One of the achievements of this work is the successful integration of two sets of two-stage belt drives and a four-bar linkage within the compact dimensions of the prosthetic design. The design challenge was to accommodate these mechanical components within a confined space without sacrificing the prosthesis’s functionality or durability. The placement of the two-stage belt drives was strategically optimized to ensure efficient power transmission while maintaining a compact profile. This configuration was vital to achieving the necessary torque and speed for the prosthesis under various walking scenarios. Incorporating a four-bar linkage into the design was a critical decision. This linkage enabled the translation of motor movements into the required ankle motions within a minimal spatial footprint. The system was fine-tuned to ensure smooth and coordinated movement, closely mirroring the natural gait cycle.

Performance testing indicated that this intricate integration did not compromise the functionality of the prosthesis. Instead, it enabled the device to provide responsive and controlled movements in both the sagittal and frontal planes, with satisfactory torque control being a notable feature for stability and comfort during walking.

### 3.2. Dynamic Responses of the System

In the step response tests, each motor drive was tested individually to verify comparable performance. A torque of 40 N·m was set as the target. The results for one actuator, depicted in Figure 14, showed a 10–90% rise time of 0.040 s and an overshoot of 16%. The other actuator exhibited a similar performance, with a rise time of 0.035 s and an overshoot of 18%, ensuring consistent functionality across both actuators.

Figure 15 displays the time domain results for the torque controller bandwidth tests. The desired torque command ($\tau_{desired}$) and the measured torque output ($\tau_{measured}$) of the actuator are plotted. For these tests, a chirp torque command sweeping from 0.1 to 40 Hz over 80 s was input to comprehensively evaluate the torque tracking performance across frequencies.

The empirical data were fit to a second-order transfer function model, which is a sensible approximation given the mass-spring-damper dynamics inherent in the physical system. Fitting the frequency response to a second-order transfer function enabled extraction of key parameters describing the dynamics, such as natural frequency, damping ratio, and bandwidth. The Bode plot for the system is shown in Figure 16.

The Bode plot analysis reveals that our system’s −3 dB bandwidth is 9.74 Hz. Comparatively, the human ankle’s torque bandwidth during walking gait is around 3.5 Hz, with torque fluctuations between 50 to 140 N·m [2]. Therefore, our system’s bandwidth sufficiently exceeds the requirements for mimicking human walking dynamics, ensuring its adequacy in replicating the necessary torque range for natural gait patterns.

### 3.3. Prosthesis Performance Evaluation

Figure 17 displays the torque tracking results achieved with the walking state machine, averaging over 50 steps. As previously mentioned, the control system primarily models ankle dynamics in the sagittal plane, treating the ankle in the frontal plane as a simple variable spring and damper, without mimicking detailed biological characteristics. This simplified approach was adopted purely for evaluating basic system performance. Appendix A demonstrates the subject walking with the state machine. The data indicate that the RMSE in the sagittal plane is 6.45 N·m, reflecting adequate capability for this specific function. Following the extensive walking trial, the zero position of the ankle remained unchanged, signifying that no tooth jumping events occurred in the gearhead during testing. This positional stability is an important indicator that the system can maintain reliability under sustained operational conditions.

## 4. Discussion

Although the achieved torque control bandwidth of 9.74 Hz is sufficient for tracking level-ground walking trajectories, it falls noticeably short of initial design expectations. The decision to omit a compliant series spring and instead rely on strain gauges for direct force-torque measurement was predicated on increasing bandwidth beyond the previous state of the art. However, extensive testing revealed that the two-stage timing belts were the dominant factor limiting dynamic performance. While the selected belts provided the strength needed for reliable torque transmission, their material compliance represents a large, unintended source of compliance relative to the other metal components. This compliance manifests as a nonlinear spring effect that not only drastically curtailed the maximum attainable bandwidth but also introduced difficulties for controller design. Specifically, the belt compliance violates the assumptions of linear dynamics that guided the original control approach. The nonlinearity distorts the frequency response away from the intended linear second-order shape, necessitating adjustments to accommodate the compliance. Furthermore, the belt dynamics vary with torque level, load, and wear over time, complicating the modeling and control design process. In summary, while satisfactory for level-ground walking, the current torque bandwidth of 9.74 Hz remains far below original targets due to the unexpected bandwidth-limiting influence of the timing belt mechanical compliance.

The torque output of the ankle could potentially be increased by incorporating a parallel spring mechanism into the system design. In this case, the carbon fiber foot could be utilized to harvest and store energy during the stance phase to assist with push-off. To enable energy storage, a virtual hard stop would need to be implemented in the controller during stance to allow elastic deformation of the foot. The stored elastic energy could then be released to contribute to propulsion in late stance, reducing actuator power requirements. Exploring the parallel spring approach is an attainable next step, as a virtual stop and release is straightforward to implement in the control software.

The weight of a prosthetic limb plays a crucial role in affecting walking stability and the overall energy expenditure of the user. Our design, characterized by its complex mechanisms and materials, inevitably adds additional weight. It is important to evaluate the potential impact of this increased weight on the user’s energy expenditure. Prior research on bionic ankle-foot prostheses has shown that such devices can normalize walking gait and reduce metabolic cost [1,2,3]. This indicates that the negative effects of added weight could be offset by improved biomechanical efficiency. Consequently, future developments of this design should focus on in-depth research to assess its effects on energy expenditure and gait symmetry, ensuring an optimal balance between weight and functionality.

Initial user feedback indicates that, while the prosthesis feels heavy at first, the user adapts quickly and appreciates the additional energy and natural feel it provides. Moving forward, design modifications and material innovations are critical in achieving an optimal balance between weight and functionality. Investigating lightweight materials, such as advanced composites or 3D-printed components, shows promise for reducing weight without compromising functional performance.

## 5. Conclusions

The 2-DoF ankle prosthesis presented demonstrates a promising initial ability to replicate natural human walking biomechanics in an untethered form factor. Benchtop frequency response and step input testing verified adequate 9.74 Hz torque control bandwidth with fast rise times, enabling tracking of biological ankle impedances up to the 3–5 Hz range typical of normal gait. Preliminary level-ground walking trials with a 77 kg subject exhibited an RMSE sagittal plane torque tracking error under 7 N·m, suggesting sufficient accuracy under body weight. Moreover, multi-step trials showed no sensor position drift, affirming the mechanical robustness of the timing belt transmission against tooth skipping under load.

While this study focused on adaptation and testing centered on sagittal plane biomechanics, the prosthesis design provides a platform for exploring coupled frontal and sagittal dynamics in future work. Specifically, the full 2-DoF architecture can enable impedance modulation in both planes to actively stabilize balance during perturbed walking, mirroring biological ankle strategies. While the current design has shown promising results, we recognize the need for ongoing enhancements. Future iterations of this prosthesis will focus on incorporating a parallel spring mechanism for improved energy storage and refining the transmission system for enhanced torque control. These improvements aim to further optimize the walking economy of individuals with lower limb amputations and expand the prosthesis’s adaptability to various terrains.

This research lays a foundation for future explorations into the coupled dynamics of frontal and sagittal plane movements. The insights gained from our work contribute to the broader goal of advancing prosthetic technology, ultimately aimed at restoring mobility and improving the quality of life for those with lower limb loss.

## Figures and Tables

**Figure 1 biomimetics-09-00076-f001:**
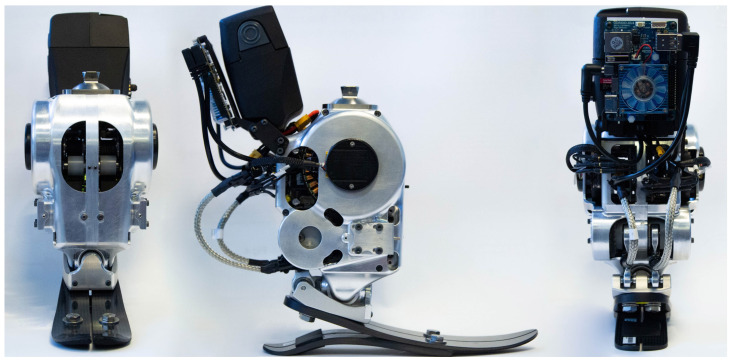
Overview of the 2-DoF ankle prosthesis, which is designed for untethered mobility, tailored for individuals around 75 kg, and suitable for varied terrains.

**Figure 2 biomimetics-09-00076-f002:**
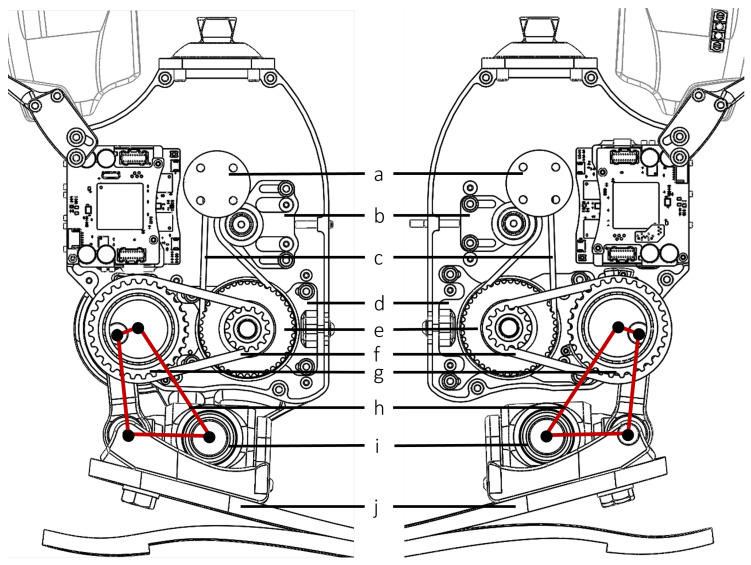
Key components in the transmission: (**a**) first-stage input pulley (12 teeth) with mounting holes for the motor; (**b**) belt tensioner for the first-stage belt; (**c**) first-stage belt (3 mm pitch, 73 teeth); (**d**) belt tensioner for the second-stage belt; (**e**) first-stage to second-stage compound pulley (44 and 13 teeth); (**f**) second-stage belt (5 mm pitch, 40 teeth); (**g**) second-stage output pulley (28 teeth), coupled with the four-bar linkage; (**h**) virtual four-bar linkage that drives the foot; (**i**) 2-DoF joint; (**j**) foot prosthesis (LP Vari-Flex foot, Össur, Frechen, Germany).

**Figure 3 biomimetics-09-00076-f003:**
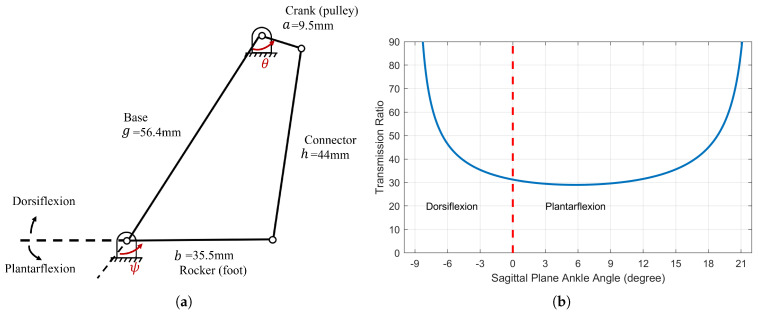
The ankle prosthesis features a transmission system that includes a four-bar linkage. This specific design choice introduces a variable transmission ratio. (**a**) The four-bar linkage in the transmission. (**b**) Overall transmission ratio versus sagittal plane ankle angle.

**Figure 4 biomimetics-09-00076-f004:**
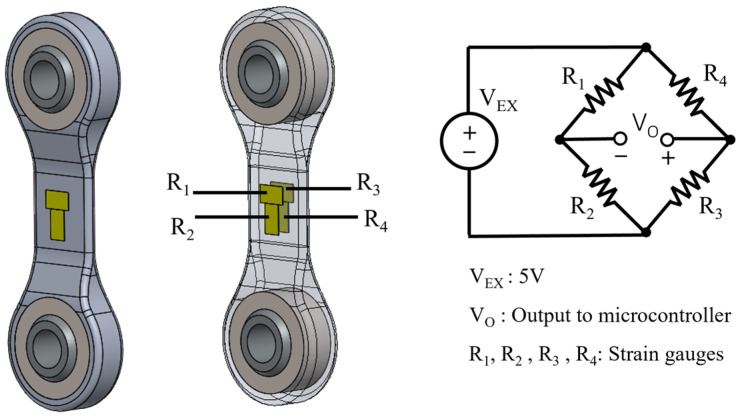
The “tendon” parts were installed with full-bridge strain gauges (half-bridge on each side) for force and torque measurements.

**Figure 5 biomimetics-09-00076-f005:**
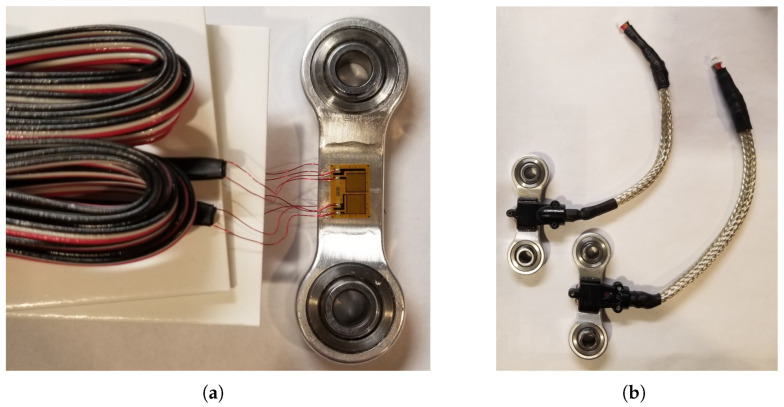
Strain gauge installation and packaging. (**a**) The pre-attached ribbon leads on the strain gauges are extremely thin (32 AWG). This serves to minimize the strain field around the gauges. (**b**) Strain gauges were packaged in 3D-printed covers, and the cables were shielded using metal-braided sleeves.

**Figure 6 biomimetics-09-00076-f006:**
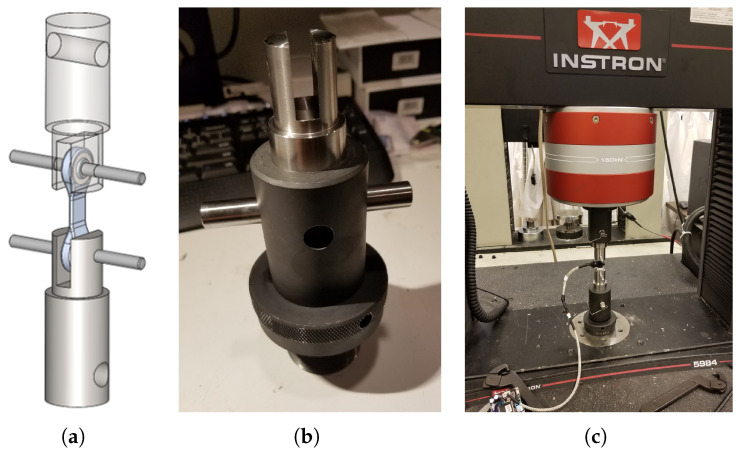
Setup for sensor calibration. (**a**) Custom fixtures were machined in 304 stainless steel. (**b**) Fitting the fixture in the universal test machine. (**c**) Test setup.

**Figure 7 biomimetics-09-00076-f007:**
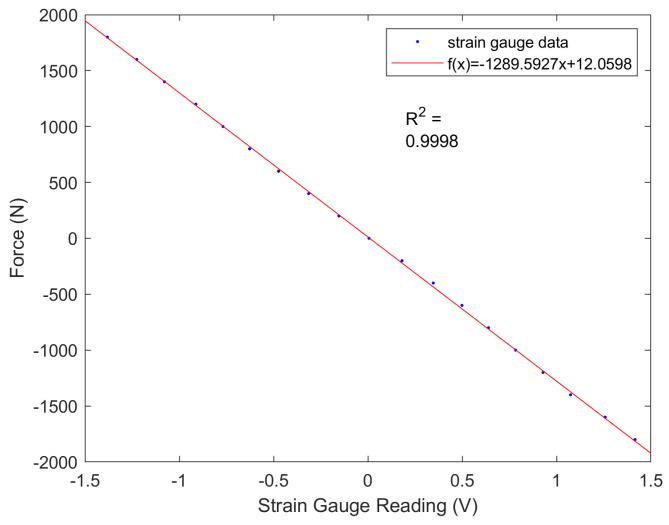
The sensor calibration and curve-fitting achieved a value of $R^{2}=0.9998$.

**Figure 8 biomimetics-09-00076-f008:**
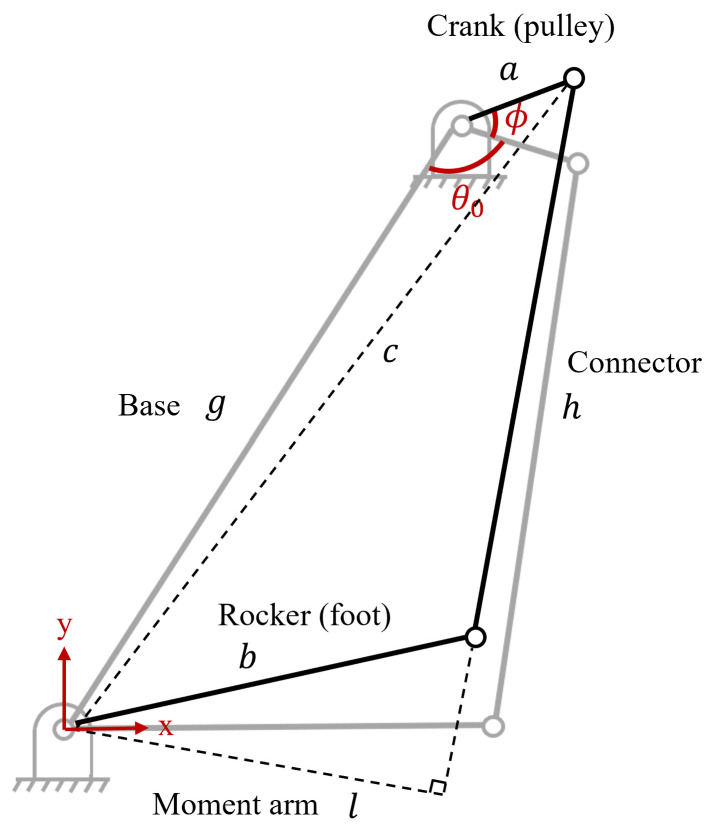
For the sagittal plane torque, the moment arm is a function of the crank angle in the four-bar linkage.

**Figure 9 biomimetics-09-00076-f009:**
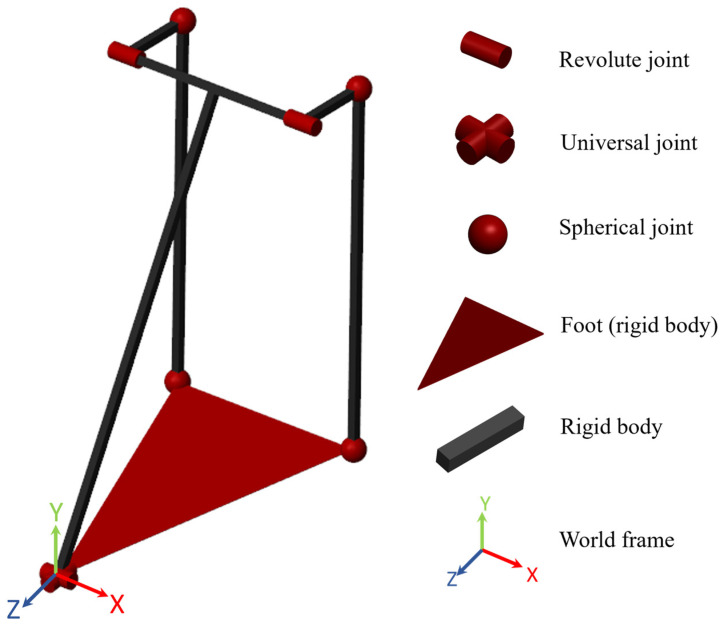
A kinematics model of the ankle was constructed using Simulink Simscape Multibody to study and map forward kinematics.

**Figure 10 biomimetics-09-00076-f010:**
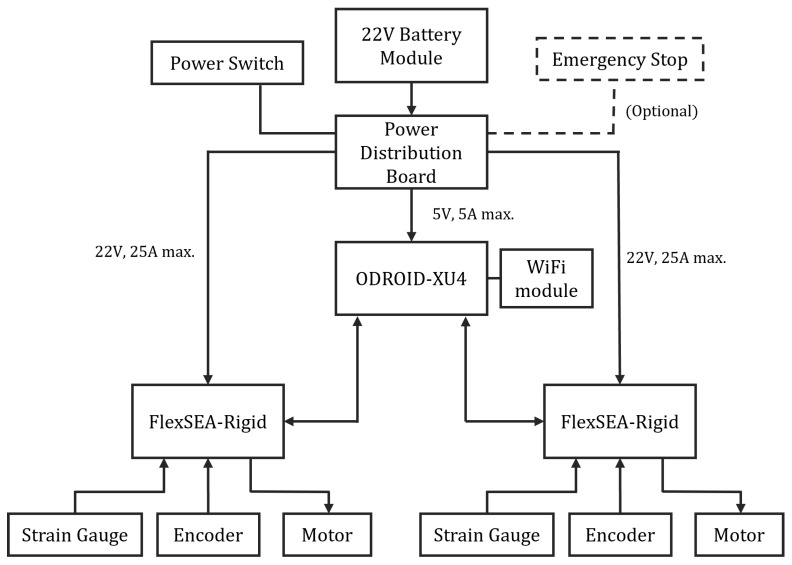
Schematics of the electrical system.

**Figure 11 biomimetics-09-00076-f011:**
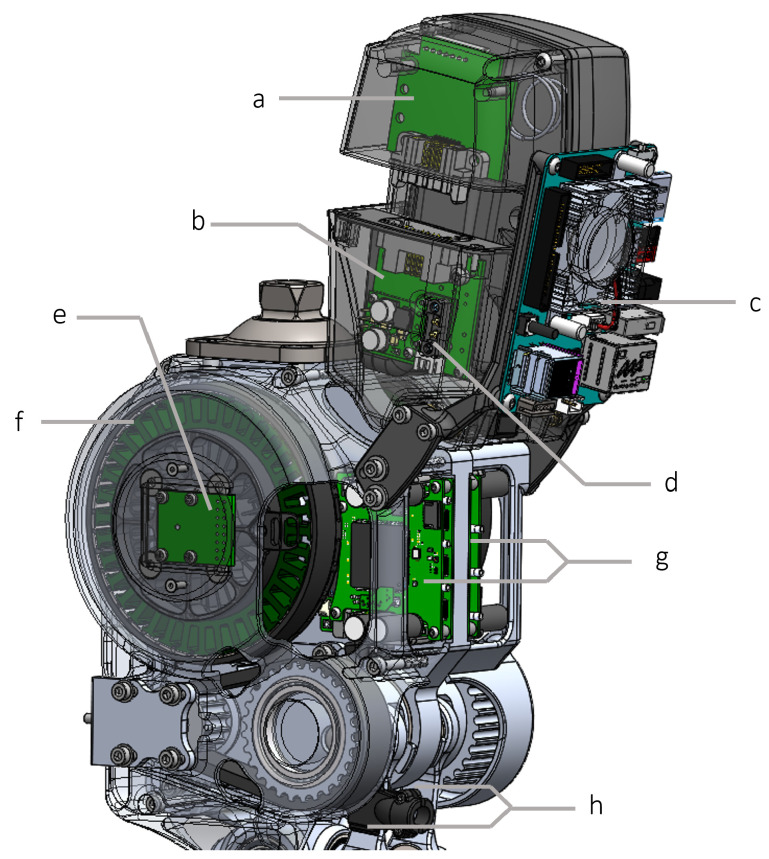
Overall electronics system: (**a**) battery module, (**b**) power distribution board, (**c**) ORDROID-XU4 single-board computer, (**d**) emergency stop connector, (**e**) absolute encoder, (**f**) motor, (**g**) FlexSEA-rigid, and (**h**) strain gauges.

**Figure 12 biomimetics-09-00076-f012:**
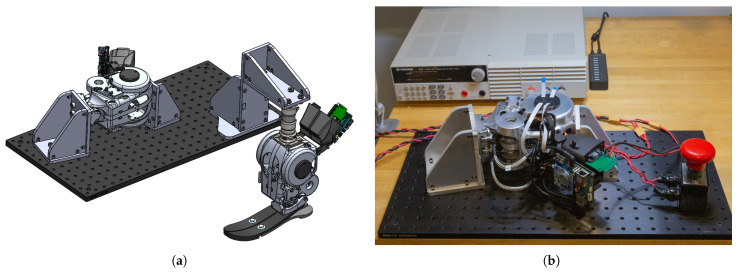
Two test fixtures were built. One is for testing and evaluating the close-loop torque controller, while the other one is for testing free space motion control. (**a**) Rendering of both fixtures. (**b**) Physical implementation of the torque-testing fixture.

**Figure 13 biomimetics-09-00076-f013:**
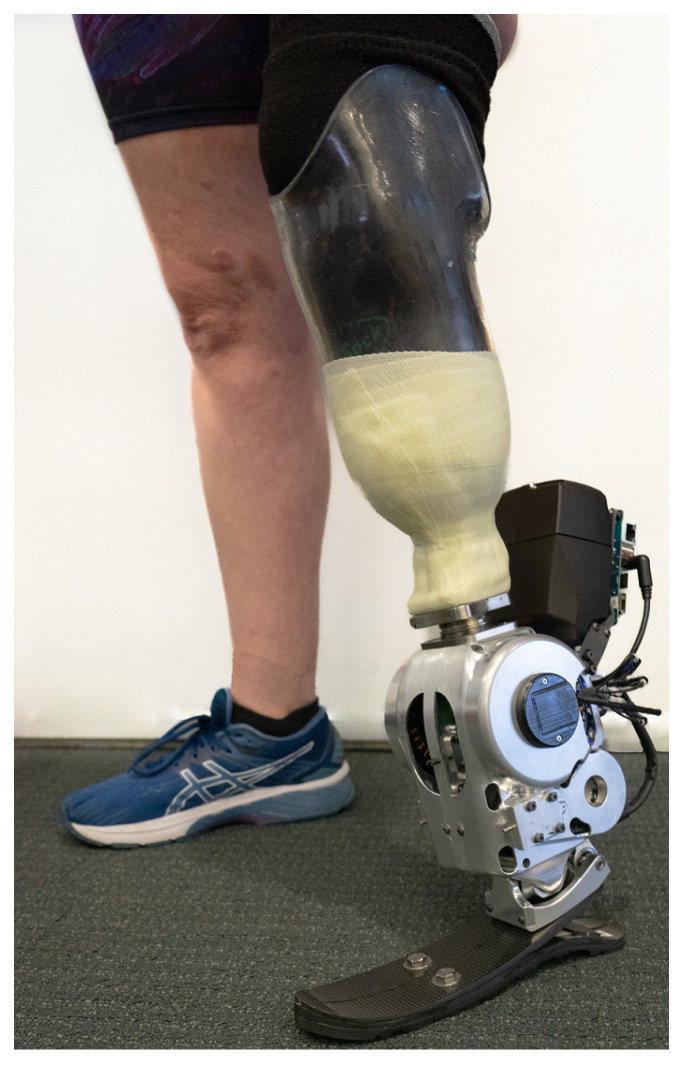
A walking trial was performed by a subject with a unilateral below-knee amputation. The goal of the trial was to evaluate torque tracking performance under body weight and to evaluate the robustness of the mechanical transmission.

**Figure 14 biomimetics-09-00076-f014:**
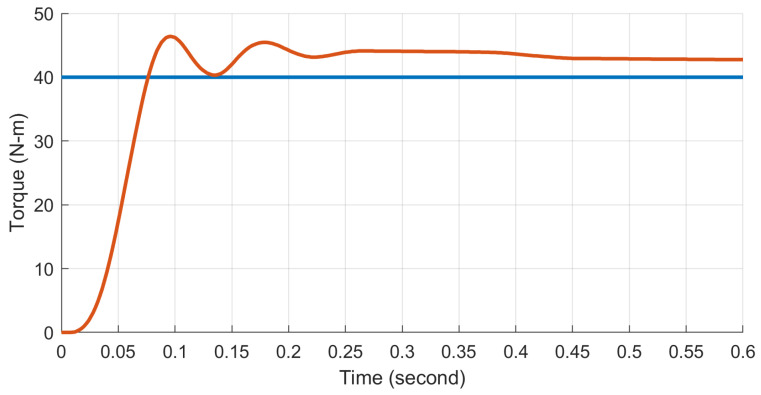
For the step response tests, a desired torque of 40 N·m was commanded.

**Figure 15 biomimetics-09-00076-f015:**
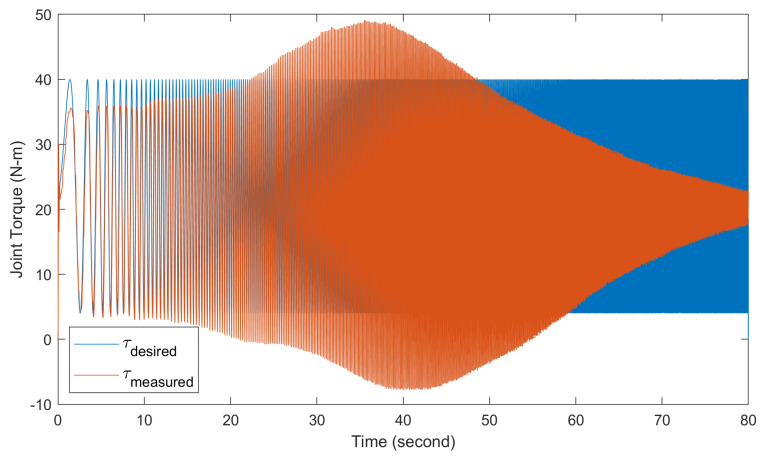
Frequency response analysis of the actuator torque control system. The figure shows the time domain torque output of the actuator in response to a chirp torque command input sweeping from 0.1 to 40 Hz over 80 s.

**Figure 16 biomimetics-09-00076-f016:**
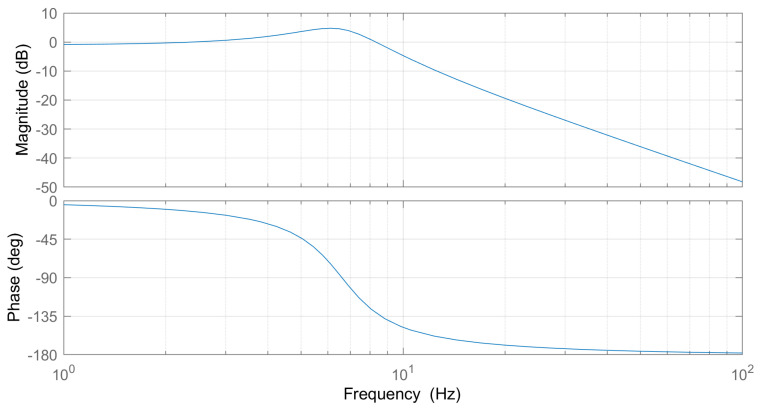
The Bode plot for the system, derived from time domain data, was modeled using a second-order system. This approach aligns with the fundamental nature of our setup, essentially a mass-spring-damper system.

**Figure 17 biomimetics-09-00076-f017:**
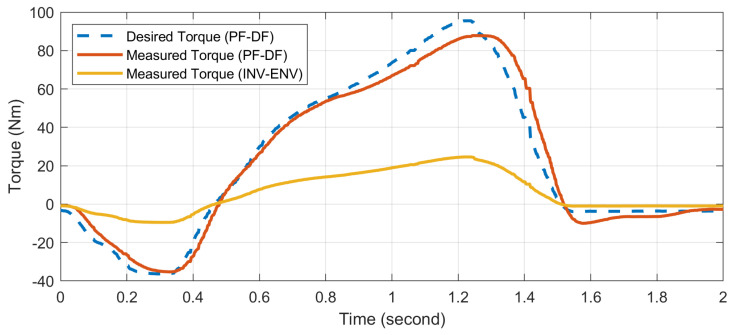
Ankle joint torque tracking performance. The figure illustrates the comparison between the desired torque trajectory, as commanded by the walking state machine, and the actual measured torque output from the ankle prosthesis, averaged over 50 steps. Performance is quantified by the RMSE of 6.45 N·m in the sagittal plane. PF-DF: Plantarflexion-Dorsiflexion; INV-EV: Inversion-Eversion.

**Table 1 biomimetics-09-00076-t001:** Design specifications.

Parameter	Value
Weight without Battery (kg)	2.42
Battery Weight (kg)	0.33
Height ^a^ (mm)	210.8
Width (mm)	126.6
Max. Allowable Inversion (deg)	13.5
Max. Allowable Eversion (deg)	13.5
Max. Allowable Dorsiflexion (deg)	9
Max. Allowable Plantarflexion (deg)	21
Transmission Ratio	32 ± 28
Peak PF-DF * Torque (N·m)	110
Peak INV-EV * Torque (N·m)	44
Battery Voltage (V)	22.2
Peak Current (A)	50
Motor Torque Constant (N·m/A)	0.088
Actuator Torque Bandwidth (Hz)	9.74

^a^ From the bottom of the foot prosthesis (without cosmesis) to the bottom of the pyramid adapter. * PF-DF: Plantarflexion-Dorsiflexion; INV-EV: Inversion-Eversion.

## Data Availability

The data related to the ankle robot discussed in this study are publicly accessible on GitHub at https://github.com/thhsieh/AT02_data (accessed on 1 December 2023).

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
