# Peer review of "Design, Characterization, and Preliminary Assessment of a Two-Degree-of-Freedom Powered Ankle–Foot Prosthesis"

_biomimetics, 2024, doi:10.3390/biomimetics9020076_

Round 1

Reviewer 1 Report

Comments and Suggestions for Authors

This manuscript is about a novel design that integrates two sets of two-stage belt drives and four-bar linkage in a small space and realizes two degrees of freedom of movement.

The material and methods of this manuscript are so detailed that it makes me consider whether some content is not necessary to present. Maybe it is better to restructure the article, I consider that the design and assembly of the prosthetic is also the result of work.

This prosthetic together with the battery weighs nearly 3 kg. Will this weight affect the stability of walking? It is recommended to have some discussions on human-machine coupling.

Reviewer 2 Report

Comments and Suggestions for Authors

biomimetics-2827545-peer-review-v1

This manuscript provides a comprehensive report on the design, characterization, and preliminary assessment of a two-degree-of-freedom powered ankle-foot prosthesis, focusing specifically on its application in an amputee participant. The manuscript is articulated and advances the current state-of-the-art powered prosthetic technology.

My primary concern pertains to the length and content of the conclusions. This could benefit from a more concise writing. I suggest that the authors consider shortening the conclusion and integrating some of its current content into the Discussion section. This restructuring enhanced the clarity and impact of the final sections of the manuscript.

Additionally, it would be highly beneficial if the authors could include or share reference videos showing the movement of participants using the prosthesis.
